# Three-Hour Argon Inhalation Has No Neuroprotective Effect after Open Traumatic Brain Injury in Rats

**DOI:** 10.3390/brainsci12070920

**Published:** 2022-07-13

**Authors:** Viktoriya V. Antonova, Denis N. Silachev, Ivan A. Ryzhkov, Konstantin N. Lapin, Sergey N. Kalabushev, Irina V. Ostrova, Lydia A. Varnakova, Oleg A. Grebenchikov

**Affiliations:** 1V.A. Negovsky Research Institute of General Reanimatology, Federal Research and Clinical Center of Intensive Care Medicine and Rehabilitology, 107031 Moscow, Russia; riamed21@gmail.com (I.A.R.); k.n.lapin@gmail.com (K.N.L.); skalabushev@fnkcrr.ru (S.N.K.); irinaostrova@mail.ru (I.V.O.); lvarnakova@fnkcrr.ru (L.A.V.); oleg.grebenchikov@yandex.ru (O.A.G.); 2A.N. Belozersky Institute of Physico-Chemical Biology, Lomonosov Moscow State University, 119992 Moscow, Russia; silachevdn@genebee.msu.ru; 3Institute of Functional Genomics, Lomonosov Moscow State University, 119991 Moscow, Russia

**Keywords:** argon, neuroprotection, traumatic brain injury, organoprotection

## Abstract

In vivo studies of the therapeutic effects of argon in traumatic brain injury (TBI) are limited, and their results are contradictory. The aim of this study was to evaluate the effect of a three-hour inhalation of argon (70%Ar/30%O_2_) after an open TBI on the severity of the neurological deficit and the degree of brain damage in rats. The experiments were performed on male Wistar rats (*n* = 35). The TBI was simulated by the dosed open brain contusion injury. The animals were divided into three groups: sham-operated (SO, *n* = 7); TBI + 70%N2/30%O_2_ (TBI, *n* = 14); TBI + 70%Ar/30%O_2_ (TBI + iAr, *n* = 14). The Neurological status was assessed over a 14-day period (using the limb-placing and cylinder tests). Magnetic resonance imaging (MRI) scans and a histological examination of the brain with an assessment of the volume of the lesions were performed 14 days after the injury. At each of the time points (days 1, 7, and 14), the limb-placing test score was lower in the TBI and TBI + iAr groups than in the SO group, while there were no significant differences between the TBI and TBI + iAr groups. Additionally, no differences were found between these groups in the cylinder test scores (day 13). The volume of brain damage (tissue loss) according to both the MRI and histological findings did not differ between the TBI and TBI + iAr groups. A three-hour inhalation of argon (70%Ar/30%O_2_) after a TBI had no neuroprotective effect.

## 1. Introduction

Traumatic brain injury (TBI) is one of the leading causes of morbidity and mortality in adults worldwide. More than 50 million cases of TBI are reported annually around the world [1]. In Russia, approximately 22,000 surgeries are performed for severe TBI every year, and the postoperative mortality remains very high, reaching 60–80% [2]. TBI ranks first among all diseases in terms of socioeconomic and medical costs [3]. Temporary and permanent disability after a traumatic brain injury is a serious public healthcare problem throughout the world. Studies of traumatic brain injury in recent decades have significantly expanded our understanding of their underlying pathophysiology and molecular mechanisms. This has resulted in the development of novel therapeutic approaches that have shown promising effects in preclinical studies and phase I/II clinical trials. However, most of these positive effects were not confirmed in phase III clinical trials [4,5]. The lack of effective clinical interventions prompts an urgent need to explore novel therapeutic targets for patients with TBI [6,7].

Recently, the attention of researchers has been drawn to noble gases such as xenon and argon due to their biological effects [8,9]. Xenon has been approved for clinical use as an anesthetic [10,11]. Argon also demonstrates anesthetic properties at elevated pressures [10]. Both argon and xenon quickly penetrate the blood–brain barrier, which explains their high speed of action [12]. Currently, there is a great body of data demonstrating neuro- and organoprotective properties of xenon [13,14,15,16]. However, argon involvement in cell protection has been studied less compared to xenon. The nephroprotective properties of argon were demonstrated by Y. Irani [17]. The neuroprotective effects of argon were first revealed in vitro on models of oxygen–glucose deprivation and TBI in organotypic mouse hippocampal section cultures [18], and further confirmed in in vivo studies using models of neonatal brain ischemia/hypoxia in rats [12,19], focal cerebral ischemia in rats [20], cardiac arrest in pigs [21], and subarachnoid hemorrhage in rats [22]. However, there are few and inconsistent data on argon efficacy in in vivo models of traumatic brain injury.

Important advantages of argon as compared to xenon are the significantly lower cost, easy transportation, no need for a special circuit, and no narcotic effect, which offers significant prospects for the clinical use of argon if evidence of its neuroprotective properties appears [23,24].

Thus, the literature review indicates a high potential for argon to protect brain neurons from ischemic damage, which emphasizes the necessity of a comprehensive study of its neuroprotective properties.

The aim of our study was to evaluate the effect of a three-hour inhalation of an argon–oxygen mixture after an open traumatic brain injury on the severity of the neurological deficit and the degree of brain damage in rats.

## 2. Materials and Methods

### 2.1. Experimental Animals

The experiments were performed on male Wistar rats weighing 275–350 g (*n* = 35). During the 12 h before the experiment, the animals did not receive food, but had free access to water. The study protocol was approved by the Local Ethics Committee of Federal Research and the Clinical Center of Intensive Care Medicine and Rehabilitology, no. 4/21/6 of 4 October 2021. The experiments were performed in accordance with the requirements of Directive 2010/63/EU of the European Parliament and of the Council of the European Union on the protection of animals used for scientific purposes.

### 2.2. TBI Simulation

Under general inhalation anesthesia with isoflurane 1.5–2.0%vol using the SomnoSuite system (Kent Scientific Corporation, USA) for low-flow anesthesia of small laboratory animals with an oxygen flow of 1 L/min, we performed a simulation of traumatic brain injury according to the method of dosed open brain contusion injury [25]. The skin on the rat’s head was shaved in the area of the operating field and treated with chlorhexidine 0.05% antiseptic. The rat was placed in a stereotactic frame, the head was secured, and a skin incision was performed along the sagittal suture. A burr hole was drilled in the parietal and frontal skull bone with a diameter of 4.5 mm with a cutter over the left hemisphere in the area of sensorimotor cortex localization, at stereotactic coordinates 2.5 mm lateral to the sagittal suture and 1.5 mm caudal to the bregma for the center of the burr hole. The trauma apparatus was placed so that the impact platform was above the dura mater. To inflict trauma, a weight of 50 g was dropped onto the platform from a height of 10 cm. The skin was sutured with Vicryl No.4 and the surgical field was treated with 5% brilliant green. The body temperature was maintained at 37 ± 0.5 °C with an electric heating pad and a rectal body temperature sensor connected to a thermoregulator until the animal recovered from anesthesia. The sham-operated animals underwent only a parietal bone depletion by 2/3 (so that the dura mater was visible) using a drill without forming a through-hole [25,26].

The animals were randomly divided into 3 groups depending on the volume of interventions performed:-Sham-operated animals undergoing anesthesia and surgery preparation without TBI + 70%N_2_/30% O_2_ inhalation (SO group), *n* = 7;-Control group with TBI + inhalation of 70%N_2_/30%O_2_ (TBI group), *n* = 14;-Experimental group with TBI + 70%Ar/O_2_30% inhalation (TBI + iAr group), *n* = 14.

### 2.3. Argon Exposure

Fifteen to thirty minutes after the injury, the animal was placed in a 15 L transparent plastic chamber continuously supplied with a fresh gas mixture (70%N_2_/30%O_2_ for the SO and TBI groups and 70%Ar/30%O_2_ for the TBI + iAr group) at a flow rate of 3 L/min. No more than 5 animals of one group remained in the chamber at a time.

The exposure time in the chamber was 3 h. After the exposure period was over, the general assessment (level of alertness, mobility) and analgesia (paracetamol 50 mg/kg subcutaneously) were performed. The animal was then moved to a cage with free access to water and food.

### 2.4. Neurological Assessment

The body weight measurement and neurological status assessment were performed the day before the experiment (D0), 24 h (D1), 7 days (D7), and 14 days (D14) after TBI induction, while the cylinder test was conducted 13 days after TBI.

The limb-placing test. The protocol based on the M. De Ryck et al. [27] technique and modified by J. Jolkkonen et al. [28] was used. Rats were hand-trained for one week before testing. The test consisted of seven tasks assessing the sensorimotor integration of the fore and hind limbs in response to tactile, proprioceptive, and visual stimulation. In the first task, the rat was held by a researcher with its forelimbs positioned on the edge of the table. Each forelimb was gently pushed off the edge one after the other and the response of the animal was registered. Normally, the rat immediately placed its limb back to its original position. In the second task, the first task was repeated with the animal’s head raised at 45 degrees so that it could not see the surface of the table or contact it with its vibrissae. In the third and fourth tasks, the rat was placed along the edge of the table and its fore- and hindlimbs were pushed aside. In the fifth test, the rat was placed on its hindlimbs on the edge of the table and then each limb was, sequentially, pushed down. In the sixth task, the rat was placed on its forelimbs near the edge of the table and gently pushed ahead towards the edge. Healthy rats resisted with their limbs, whereas rats with TBI did not resist, and their limbs slipped off the edge of the table. In the seventh task, the rat was held at the base of its tail and slowly lowered to the surface of the table. Normally, the rat stretched its forelimbs forward to approximately 10 cm to the surface of the table. Each task was scored as follows: normal performance, 2 points; delayed (>2 s) and/or incomplete performance, 1 point; failure to perform the test, 0 points. Scores were summed and results were presented as the sum of points for the test.

The limb-use asymmetry (cylinder) test. In this test, the number of forelimb placements on the wall performed by the animal separately using the impaired forelimb, unimpaired forelimb, and both limbs was counted, and then the percentage of placements of the forelimb out of the total number of placements was calculated. The higher the percentage of the placements of the impaired limb, the greater the recovery of nervous system functions after the injury [29].

### 2.5. MRI Scan

On day 14 after the traumatic brain injury, the animals were examined using MRI. Imaging was performed on a tomograph with a magnetic field induction of 7 Tesla and a gradient system of 105 mTl/m (BioSpec 70/30, Bruker, Munich, Germany). The animals were anesthetized with isoflurane (1.5–2%) and placed in a positioning device with stereotaxis and a thermoregulation system.

A standard protocol for rat brain examination was used, which included T2-weighted imaging. A linear transmitter with an internal diameter of 72 mm was used for the radiofrequency (RF) signal transmission, and a surface receiving coil for the rat brains was used for the RF signal detection. The following pulse sequences (PS) were used: RARE, a spin echo-based PS with the following parameters: TR = 6000 ms; TE = 63.9 ms; 0.8 mm slice thickness in 0.8 mm increments, 256–384 matrix size, and 0.164–0.164 mm/pixel resolution. The total scanning time per animal was approximately 25 min. The degree of brain damage was assessed using a graphical analysis of MR images by calculating the volume of the damaged brain area. For this purpose, a slide with the largest brain lesion area was selected in a series of MR images. ImageJ software (National Institutes of Health image software, Bethesda, MD, USA) was used to calculate the lesion area in mm^2^. Then, the brain lesion area was similarly calculated on four more slides (two cranial and two caudal). The volume of the brain injury was calculated using the following formula:V = ΣSn × d
where d is the thickness of one slice (0.8 mm), and ΣSn is the sum of the lesion areas on five slides (mm^2^) [26].

### 2.6. Histological Examination

For the histological examination (determination of lesion volume), the anterior cortex was extracted from the rats immediately after euthanasia (decapitation under sevoflurane anesthesia), and fixed for 48 h in 10% buffered formalin (Biovitrum, Russia) on day 14 after the trauma. The material was subjected to routine processing and was embedded in paraffin. Serial brain sections of 5–6 µm thickness with a step of 500 µm were created on a rotary microtome. Micropreparations were stained with Nissl cresyl violet. We analyzed 5 serial brain sections with a maximum brain tissue damage area (at the 3.15 ± 2.45 Bregma level according to the rat brain atlas) [30]. Using a ScanScope CS digital scanner (Leica Biosystems, Vista, CA, USA), we obtained images of brain sections. Using the NIS-Elements BR software (Nikon Corp., Tokyo, Japan), the external borders of visually undamaged tissue and ventricles were outlined on the image in each hemisphere and their area was measured. The area (S) of damage was calculated using the following formula:Tissue loss per section = (S of the contralateral hemisphere − S of the ventricles of the contralateral hemisphere) − (S of the ipsilateral hemisphere − S of the ventricles of the ipsilateral hemisphere).

The obtained number was multiplied by the section thickness (6 μm) and summed up to estimate the volume of tissue loss (V of tissue loss (divot)) for each rat. The volume (V) of brain damage in the ipsilateral hemisphere was expressed as a percentage of the brain tissue volume of the contralateral hemisphere using the following formula [31]:%V = (V tissue loss/V contralateral hemisphere) × 100%.

Mortality in the animal groups was assessed on days 1, 7, and 14 after the trauma.

### 2.7. Statistical Analysis

The data were analyzed using STATISTICA 7.0 (StatSoft. Inc., Tulsa, OK, USA) and GraphPad Prizm software. The normality of data distribution in the samples was assessed using the Shapiro–Wilk criterion. All data were presented as median, quartiles, and single values. Statistical differences in data with at least one group having a non-normal distribution were analyzed using the Kruskal–Wallis test with Dunn’s multiple comparison or Mann–Whitney U-test with Bonferroni correction for the comparison of three or more groups or the Mann–Whitney U-test for the analysis of two groups. The differences were considered significant at *p* < 0.05.

## 3. Results

During the 14 days of the experiment, nine animals were withdrawn due to reaching the humane endpoints (four animals from the TBI group and five animals from the TBI + iAr group), two animals were excluded from the group due to a developed abscess, and two animals were excluded because the amount of injury on their MRI and brain histology was extremely large, and so, they were not included in the ensuing analysis.

### 3.1. Neurological Assessment and Body Mass Analysis

Before the start of the experiments, at the D0 time point, the groups did not differ in terms of animal weight (Figure 1a). The analysis of body weight at different time points after TBI also revealed no significant differences between the groups at points D1, D7, and D14 (Figure 1b–d).

The analysis of weight changes within the groups showed a similar trend (Figure 1f,g); at the D1 point, animal weight in all groups decreased compared to D0 (in the SO group—significantly) and then recovered postsurgery. The rate of body weight recovery in the experimental groups differed. In the SO group, animal weight returned to the preoperative level at D7 and remained at the same level until D14. In the TBI group, animal weight significantly increased at D14 compared to D0. In the TBI + iAr group, the animal weights at D0, D1, D7, and D14 did not differ.

The limb-placing test (LPT): at each of the time points (D1, D7, and D14), the sum of the LPT scores in the animals of both experimental groups was lower than that in the SO group, and the TBI and TBI + iAr groups did not differ from each other (Figure 2a–c).

The values of this parameter did not change over time from D1 to D14 in the SO group of animals (Figure 2d). At point D14, the total LPT score in the TBI group was significantly higher not only compared to D1 (*p* = 0.0039), but also compared to D7 (*p* = 0.0156). In the TBI + iAr group, the assessment scores at D1, D7, and D14 did not differ.

The limb-use asymmetry (cylinder) test: The percentages of using the impaired forelimb for the SO group were averaged 20 ± 17%, for the TBI group were averaged 5 ± 8.7%, and for the TBI + iAr group were averaged 2 ± 4.1% (Figure 3). Compared to the SO group, the quantity of using the impaired limb in the TBI and TBI + iAr groups was lower (*p* = 0.0484 and *p* = 0.0492, respectively).

### 3.2. Brain MRI

The mean lesion volume (Figure 4c) in the TBI and TBI + iAr groups was 34.4 ± 11.6 mm^3^ and 30.3 ± 10.6 mm^3^, respectively, without significant differences between the groups (*p* = 0.8591). The example of an MRI image can be seen in Figure 4a.

### 3.3. Histological Examination of the Brain

All of the animals who suffered a traumatic brain injury showed varying degrees of ventricular dilatation and cortical damage in the ipsilateral cerebral hemisphere (all together considered as tissue loss) on T2-weighted MR-images 14 days after TBI induction (Figure 4a). A similar pattern was observed on histological slices (Figure 4b). According to the histological data, tissue loss in the TBI group averaged 25.3 [17.8; 28.2]% of the volume of the contralateral undamaged hemisphere, whereas, in the TBI + iAr group, tissue loss was 22.3 [12.3; 23.3]%. The argon treatment had no significant effect on the total amount of brain tissue injury caused by TBI (Ltqcnd) (Figure 4d).

## 4. Discussion

The major finding of our research was the lack of a neuroprotective effect of the three-hour argon inhalation used in the first hour after an open traumatic brain injury in rats. This was indicated both by the results of a functional assessment of a neurological deficit during the 14 days of the post-traumatic period and by the size of the lesion according to the living MRI and histological examination of the brain on day 14 after the trauma. The duration of the experiment could be considered sufficient for the delayed therapeutic effects of the drug to appear.

Currently, early preclinical studies of argon’s potential therapeutic effects in ischemic reperfusion and traumatic brain injury are underway. Another noble gas, xenon, in contrast to argon, has already been approved for clinical use as a general anesthetic and its neuroprotective properties in many in vitro and in vivo studies [11,13,14,15,16,32] have been confirmed. However, its use in routine clinical practice is challenging due to its high cost, as well as narcotic effect, which complicates the assessment of patients’ neurological status [5]. Unlike xenon, argon is inexpensive and does not require a closed breathing circuit. Argon has no sedative properties, and therefore, does not affect the neurological status. A simple administration (through a face mask) and lack of toxicity would allow starting its use at the earliest possible time from the moment of the patient’s admission [20]. In a clinical study involving cardiac surgery patients, a 15 min argon inhalation (70%Ar/30%O2) was shown to have no effect on the cerebral blood flow and oxygen and glucose metabolism [33]. In addition, argon has no cerebral vasodilating effects, which is a key limiting factor in patients with TBI at the risk of intracranial hypertension [6]. Therefore, the use of argon, if proven to have neuroprotective properties, could significantly improve clinical outcomes of cerebral accidents.

Argon is considered to be a chemically inert substance, but it exhibits biological effects. In vitro and in vivo studies on different models of hypoxic/ischemic lesions [4,19,20,21,22,34,35] have shown a good neuroprotective potential of argon. Thus, argon inhalation caused a decrease in neuronal death [5,22,35], and an infarct volume in the focal cerebral ischemia model [20]. Moreover, a 24 h exposure to 70%Ar/30%O2 [12,36] resulted in a reduced severity of the neurological deficit [4,36] in the postischemic period in stroke models.

The results of studies of the neuroprotective effects of argon are often contradictory [6,20,23,35]. For example, in mice, argon inhalation for 24 h after TBI resulted in a decrease in the neurological deficit, improvement of cognitive functions in behavioral tests, and a reduction in cerebral white matter edema and damage [6]. At the same time, a study with a similar design failed to reveal the positive effects of argon [23]. Other authors pointed out that the efficiency of argon action varied depending on the specific experimental conditions (permanent or transient ischemia, different times of argon exposure) [20], as well as on brain regions [37]. In contrast to xenon, the molecular mechanisms of argon action are still poorly understood. Xenon is known to be an antagonist of the N-methyl-D-aspartic acid receptor, activating adenosine-5′-triphosphate-sensitive potassium channels and two-pore potassium channels [12]. The exact mechanisms of the positive effects of argon are still far from being clear. Apparently, this gas, unlike xenon, has other molecular targets [35]. Argon has been found to affect ɣ-aminobutyric acid type A receptors, although it is not clear whether this is related to cytoprotection [12]. In the in vitro and in vivo models of hypoxia, argon was demonstrated to cause the activation of several proteins, including the Nrf2 nuclear factor, NAD(P)H dehydrogenase (NQO1), and SOD1 superoxide dismutase, which caused a decrease in oxidative stress, neuroinflammation, and, consequently, reduced neuronal death [19]. The positive effect of argon can be associated with an increased expression of HIF-1 (hypoxia-inducible factor 1α), HO-1 (heme oxygenase 1) [22,38,39], and antiapoptotic protein Bcl-2 [12], with the inhibition of the GSK-3β (glycogen synthase kinase-3β) enzyme [19], and the increased activity of ERK 1/2a (extracellular-regulated kinase 1/2a), an enzyme involved in the processes of cell proliferation and survival, probably due to the direct activation of MEK (mitogen-activated extracellular signal-regulated kinase) [19,39,40]. The Raf/MEK/ERK MAP kinase pathway is known to promote cell survival by blocking NF-kB, which leads to the enhanced transcription of Bcl-2, Mcl-1, and antiapoptotic genes [41].

The lack of a positive effect of argon in our study can be linked to the specific design of our experiment, particularly to the rather short time of argon exposure (3 h). In several studies, the argon exposure was 24 h [6,20,23]. Other factors that may affect the manifestation of the neuroprotective properties of argon are the mode of its administration (before or after the onset of the pathological process), as well as its fractional concentration in the gas mixture [42]. In addition, the neuroprotective effect of argon was assessed based on morphological criteria, while the molecular markers of brain damage in TBI were not investigated. A better understanding of the molecular mechanisms of argon effects in TBI would enable a more rational selection of the experimental model and the endpoints of the study.

Thus, conflicting data regarding the neuroprotective properties of argon indicated the need for a systematic review (meta-analysis) of preclinical studies on this topic with a comparison of the different treatment protocols. This would make it possible to reasonably choose the most appropriate experimental model, as well as to determine the most promising direction when planning randomized clinical trials in this area.

## 5. Conclusions

The three-hour inhalation of an argon–oxygen mixture (Ar 70%/O2 30%) after traumatic brain injury had no neuroprotective effect; in particular, it neither reduced the volume of brain lesions nor decreased the severity of the neurological deficit.

## Figures and Tables

**Figure 1 brainsci-12-00920-f001:**
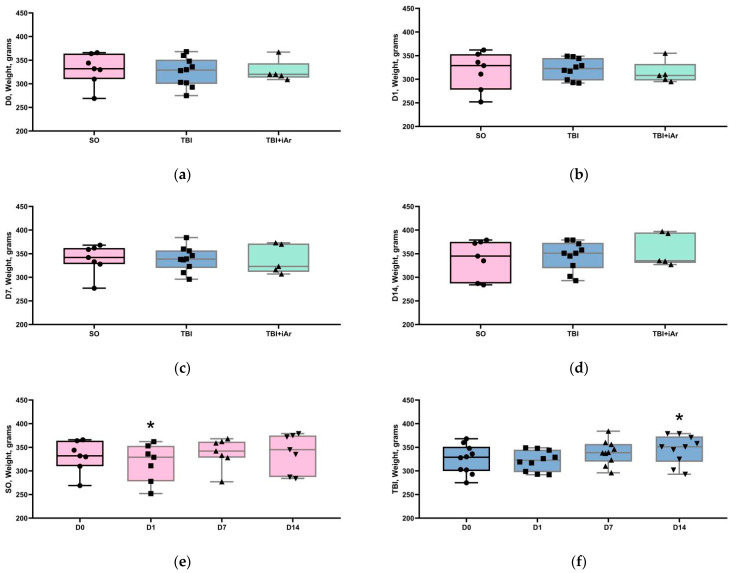
Analysis of body weight data. (**a**) Weight of rats the day before the simulated TBI; (**b**) weight of rats on the day of the simulated TBI; (**c**) weight of rats 7 days after the simulated TBI; (**d**) weight of rats 14 days after the simulated TBI; (**e**) weight change in the control group; (f) weight change in the TBI group; (**g**) weight change in the TBI + iAr group. The data are presented as median, quartiles, and single values. *—*p* < 0.05 vs. D0, Kruskal–Wallis test with Dunn’s multiple comparisons. ■; ●; ▲; ▼—single values within specific statistical sample.

**Figure 2 brainsci-12-00920-f002:**
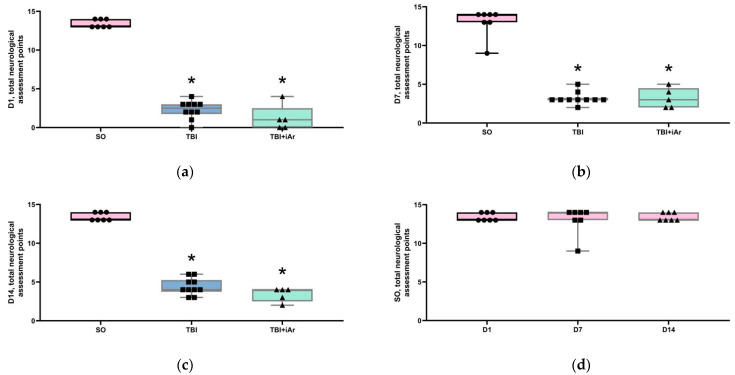
(**a**) LPT score 1 day after the simulation of TBI; (**b**) LPT score 7 days after the simulation of TBI; (**c**) LPT score 14 days after the simulation of TBI; (**d**) changes in LPT score in the control group; (**e**) changes in LPT score in the TBI group; (**f**) changes in LPT score in the TBI + iAr group. The data are presented as median, quartiles, and single values. *—*p* < 0.05 vs. SO group on (**a**–**c**) figures, vs. D1 group on (**d**–**f**) figures; #—*p* < 0.05 vs. D7 group, Kruskal–Wallis test with Dunn’s multiple comparisons. ■; ●; ▲—single values within specific statistical sample.

**Figure 3 brainsci-12-00920-f003:**
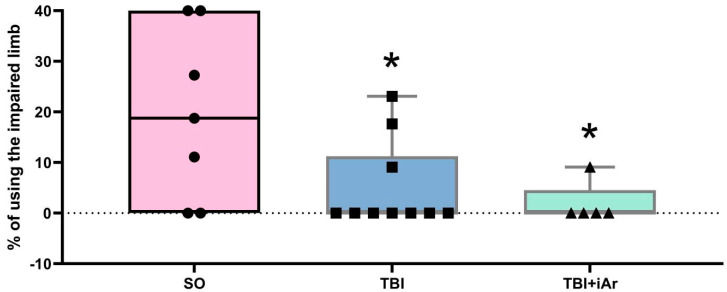
The cylinder test results. Day 13 day after the traumatic brain injury. The data are presented as median, quartiles, and single values. *—*p* < 0.05 vs. SO group, Mann–Whitney U-test with Bonferroni correction. ■; ●; ▲—single values within specific statistical sample.

**Figure 4 brainsci-12-00920-f004:**
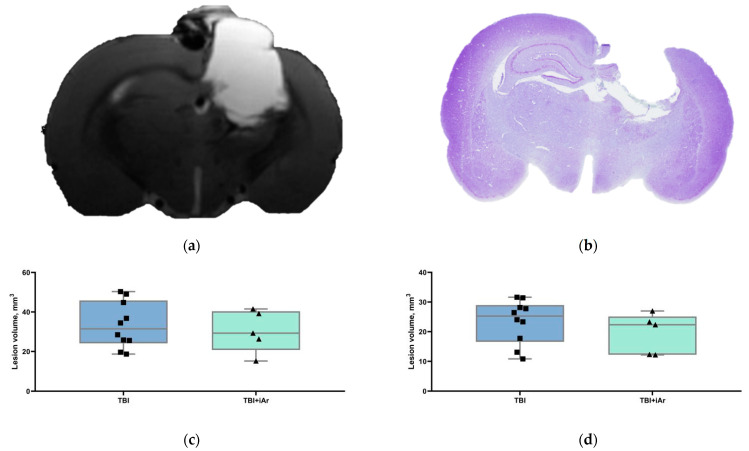
(**a**) Representative T2-weighted coronal MR-images and (**b**) cresyl violet-stained sections of the rat brain on day 14 post-TBI demonstrating loss of cortical tissue at the site of impact. (**c**) Volume of brain lesions caused by traumatic brain injury in the TBI and TBI + iAr groups based on MRI data. (**d**) Volume of brain lesions caused by traumatic brain injury in the TBI and TBI + iAr groups based on histological examination data. No significant difference between the groups was found (Mann–Whitney U criterion). The data are presented as median, quartiles, and single values. ■; ▲—single values within specific statistical sample.

## Data Availability

Not applicable.

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
