# Peer review of "Three-Hour Argon Inhalation Has No Neuroprotective Effect after Open Traumatic Brain Injury in Rats"

_brainsci, 2022, doi:10.3390/brainsci12070920_

Round 1
Reviewer 1 Report
The authors demonstrate that argon inhalation immediately post TBI does not improve neurological outcome.
This is a solid and well-written manuscript. The study designs are clear. All of the essential experimental details are provided. The results are clear. The Introduction, Discussion, and conclusions are of appropriate length and content.
The authors used several assays of TBI injury including neurological scoring at 1, 7, and 14 days using a limb placement test, a limb use test, MRI, and histological exam by cresol violet staining. All of these tests showed essentially the same result: As compared to sham controls, the TBI animals evidenced significant brain damage. This damage was not mitigated by treatment with argon inhalation therapy. I found the data convincing, in large measure because the methods were well-described and left no ambiguity as to what was done, how the data was analyzed, and how the results were interpreted. Statistical treatment was fine. Authors mentioned mortality and exclusion criteria. The discussion was suitably conservative and the authors did not make claims beyond what their data suggested. I could not find anything to negatively critique.
This was a straight-forward study with clearly described experimental methods and results. As I indicated in my Comments to Editors, the only ostensible issue was this was a negative study, meaning there was no effect of the Ar treatment on TBI. I argued this is not an issue, and that, given the strength of the study, the negative result is clear-cut. Because there are people researching the use of noble gases as a possible TBI treatment, the work is timely and relevant to this research area.
Reviewer 2 Report
The authors tested a 3h treatment of 70% argon in 30%oxygen for its neuroprotective capacity in an open TBI model in rats by MRI and histological analysis after 14 days and body weight and neurological testing at d1, d7, and d14. They found no significant protective effect by the treatment.
The following points may be considered:
Methods:
- Anesthesia was performed with isoflurane only - how was the analgesia ensured with Isoflurane having no analgesic effect? Minor point: Page 2 line 77: concentration of Isoflurane 1.5-2 0/00 - you may probably mean %?
- The model may be descibed with slighlty more detail: what is the exact size of the round (?) craniotomy, are the given stereotactic ccordinates defining the midpoint of the circle? Parietal bone depletion in sham operated animals: what does this exactly mean - was the bone only thinned without removal?
- Paracetamol: may not be named anesthesia, but analgesia (page 3, line 107): dosage may be quite low (for mice, 200mg/kg is recommended for s.c. injection, with reinjection every 2-4 hours, for rats mainly oral application (in drinking water, contiuous application over few days) is recommended - please comment on the appropriateness of the comparable low dosage and only one application and how pain assessment was performed.
- limb placing test: please provide some more information of the seven test-parts - it is a bit unconvenient to search within the references to get an idea. The term "trial" may be missleading here, it may be interpreted as a 7x repetition and not seven different subtests...
- MRI scan: why as the imaging only performed at the end of the observation period? The lesion development may have been documented over time by this non-invasive imaging method to investigate whether argon may have slowed lesion maturation.
Statistics:
- "The data are presented as mean ± standard deviation" - I did not find any such data presentation - all data are presented in the much more appropriate way of "median, quartiles and single values", please correct this on page 4,line 174-175.
- The statistical analysis is not correct: Mann-Whitney test is only appropriate to compare two groups. For comparison of three or more groups, the Kruskal-Wallis test followed by posthoc tests has to be used. It is not appropriate to perform several Mann-Whitney (or t) tests, comparing two groups at a time, even with posthoc correction.
- In principle, the data of this study may be more correctly analysed by 2-Way-ANOVA - with time as one factor and treatment (sham, TBI, TBI+Arg) as the second factor. With those small sample sizes, it may not be such relevant that some of the data are not normally distributed.
Results:
- MRI lesion volumes: two animals with extremely large volumes: what is the reason for these outliers? Is it an artifact of MRI imaging, were the lesions of these two animals comparably large in the histological analysis? If yes, it is recommended to exclude these two animals from the whole analysis, and their data should also be removed from the behavior and body weight analysis. Please commend on this.
- Minor points: figure legend of fig 1 and 2: the part "The data are presented as mean ± standard deviation" should be deleted
Discussion:
- A very recently published paper using Argon treatment at different concentrations for 4h after CCI in mice with 24h endpoint is not incuded in the discussion: Schneider et al., Biology, 2022.
doi:10.3390/biology11020158
- While discussing the lack of effect seen in their study, the authors focus mainly on the short exposure of Argon for 3h - another reason may be a too high concentration - a more detailed discussion on the amount of argon used may be helpful - 25% or 50% may have been better than 70%?
- In the conclusion, the authors mention the need for more preclinical studies. In my view, I would much more recommend to at first perform a systematic review and metanalysis with comparison of the different treatment protocols (% argon, duration, start of teatment, disease model, species) and from this develop the most promosing protocol for a multicenter study, to avoid the use of any more animal with only slightly modified protocols in a single center approach.
Round 2
Reviewer 2 Report
all points have been adequately addressed in the revised manuscript.